# Learning outcomes of structured perioperative teaching based on adult learning

**Nan-Chieh Chen**[1], **Yu-Tang Chang**[2,3], **Po-Chih Chang**[3,4], **Cheng-Sheng Chen**[5,6]*, **Chung-Sheng Lai**[3,7]*

**1** Department of Medical Humanities and Education, College of Medicine, Kaohsiung Medical University, Kaohsiung, Taiwan, **2** Division of Pediatric Surgery, Department of Surgery, Kaohsiung Medical University Hospital, Kaohsiung, Taiwan, **3** Department of Surgery, College of Medicine, Kaohsiung Medical University, Kaohsiung, Taiwan, **4** Division of Thoracic Surgery, Department of Surgery, Kaohsiung Medical University Hospital, Kaohsiung, Taiwan, **5** Department of Psychiatry, College of Medicine, Kaohsiung Medical University, Kaohsiung, Taiwan, **6** Department of Psychiatry, Kaohsiung Medical University Hospital, Kaohsiung, Taiwan, **7** Division of Plastic Surgery, Department of Surgery, Kaohsiung Medical University Hospital, Kaohsiung, Taiwan

* sheng@kmu.edu.tw (CSC); chshla@kmu.edu.tw (CSL)

**Data Availability Statement:** All relevant data are within the manuscript and its Supporting Information files.

**Funding:** The author(s) received no specific funding for this work.

## Abstract

### Background

Self-directed learning is the cornerstone of adult learning. The aim of the study was to investigate the improvement of core competency and increase interest to be a surgeon among medical students after a perioperative training through a structured learning with written record model. The mediating role of adult learning pattern on core competency was also examined.

### Methods

A 2-week training protocol was based on a structural learning model which included a structured written record by the learner for postoperative immediate feedback. An adult learning questionnaire (ALQ) was developed to assess learners' adult learning pattern and a clinical core competency questionnaire (CCCQ) was developed to assess learning outcomes. A two-way repeated measured of ANCOVA would be used to analyze the interaction effect of adult learning pattern and learning effect on learning outcomes.

### Results

From Jan 2017 to Dec 2019, 412 medical students were enrolled in the study. The increase scores of CCCQ and a significant numbers of increase interest to be a surgeon were shown after the perioperative training. Two-way repeated measure ANOVA revealed that there were significant differences in change between pre- and post-CCCQ across four levels of ALQ (interaction effect F = 13.0, p <0.001). The more adult learning patterns medical students own, the more they will benefit from the training.

**Competing interests:** The authors have declared that no competing interests exist.

**Abbreviations:** BID, Briefing, Intraoperative teaching, and Debriefing; ALQ, Adult learning questionnaire; CCCQ, Clinical core competency questionnaire; ACGME, Accreditation Council for Graduate Medical Education; PR, percentile rank.

## Conclusions

The structural learning with written record model provides an effective perioperative training represented with clinical core competency and increase the interest to be a surgeon in the future. Medical students with tendency of adult learning pattern would learn better.

## 1. Introduction

Nowadays, recruitment of surgical resident physicians has been challenging in Taiwan, and it is not unusual that teaching hospitals encounter difficulties in recruiting adequate numbers of surgical physicians. This problem can be attributed to one of the factors being the extremely heavy work load, and two, the shortage of systemic didactics in perioperative training [1, 2]. Effective perioperative training should provide students with more information about knowledge and skills for operating so that more interest and intention to being surgeons might be aroused. Both these factors can affect career choice for young physicians. However, to improve the working environment by reducing working hours might lead to inadequate training in the operating room. It is necessary to develop an effective learning environment to balance learning and teaching effectiveness and working load. More effort is required to attract and recruit more young physicians through effective surgical education and training.

Perioperative training is one of the most important components of surgical education and training [3]. Traditionally, surgical education is proceeded using an apprenticeship model [4], where the learner acts as an apprentice to the qualified surgeon, observing, studying, and eventually participating in real cases. Currently, surgical education is shifting toward an emphasis on teaching and learning efficiency in an operating room, in where trainees can reap the greatest benefits from every clinical opportunity by optimizing teaching [4]. To maximize the learning effectiveness for surgical trainees in the operating room, teaching and feedback are of critical importance. However, previous studies have suggested a disparity might exist between learners and teaching faculty in terms of perioperative teaching, including preoperative preparation, intraoperative and postoperative feedback [5–8]. Therefore, how learners make judgements and receive feedback about perioperative teaching and learning should be also taken into consideration, not only the faculty staff. The learning effect is decided by the attitude and learning strategies of the students themselves [9]. The tight pace in the operating room is an undoubted challenge for perioperative training, because it must be implemented in a very short period [8, 10, 11]. Therefore, effective teaching and learning through a structured learning model is of paramount importance.

Adult Learning theory is to develop skills for self-directed lifelong learning. The theory points out that adults learn best when they know why they need to learn something, they can use self-directed learning, the learning involves real-life situations, and the stimulus for learning is internal rather than external [12]. Adult Learning Theory has been developed for over two decades, and has been applied in the field of medical education as well [13], including surgical education [14]. Development of core competencies for surgeons have relied on education in the operating theatre [15]. Adult learning principles have been perceived that could be employed regularly to improve surgical training [14].

Several measurements for learning effect of medical education have been proposed. The Accreditation Council for Graduate Medical Education (ACGME) developed a competency program in 1999 [16]. The goal of this program was to demonstrate successful achievement of well-defined clinical education outcomes. Six competencies, namely patient care, medical

knowledge, practice-based learning and improvement, interpersonal and communication skills, medical professionalism, as well as systems-based practice, have been put forth [16].

In Taiwan, medical clerkship rotation in each surgical sub-specialist division usually takes a short period of around two weeks, and to enhance their learning performance has effectively become a very important issue. This study was conducted to evaluate the learning effect of medical students undergoing their structural perioperative training, and aimed to clarify the triangular relationships between clinical core competencies, adult learning competency, and how the interest toward being surgeons increased.

## 2. Materials and methods

### 2.1. Structured learning model with written record

This study was conducted in an university teaching hospital in Taiwan from Jan 2017 to Dec 2019. Medical students experiencing their 2-week rotation in the Division of Plastic Surgery, Pediatric Surgery, and Thoracic Surgery were enrolled. The research protocol was approved by the Institutional Review Board.

A structural learning with written record model was developed through experts consensus meeting. The model involves reinforcement, correction, and generation of rules to guide future practice. The research team developed a 2-week learner-centered training protocol, including three sections of briefing, intraoperative teaching, and debriefing [10]. Faculty discussed the basic knowledge of surgery-related living anatomy, procedure, disease course, and possible outcome at the briefing section, while during the intraoperative teaching section, interaction between faculty members and learners occurred in several ways, such as real-time feedback, then during the debriefing section following surgery. Learner would be required to complete a perioperative written record, which included sections of diagnosis, surgical indication, type of operation, anatomy involved, postoperative care of the patient undergoing surgery. The written record functioned as a reflection of participation in the surgery in terms of medical knowledge, interaction with faculty, learning experience, and effects of self-learning. The intention was that the written record would allow learners to become aware of adult learning principles, then the teacher would closely review the written record and give direct feedback to the learners immediately after operation. Application of the problem-centered Questions and Answers (Q &A) interaction related to the patients' condition could evoke the awareness of learning needs and learners' clinical reasoning [11]. The structural learning model with written record designed in this study was based on adult learning pattern and clinical reasoning of teaching and learning.

### 2.2. Assessment

For the Adult Learning Questionnaire (ALQ), our research team developed a 6-item questionnaire to assess the adult learning pattern of the learners. The constructs of the ALQ were learner experience, self-directed learning, readiness to learn, orientation to learn, motivation to learn, and need to know [17–20]. Six items are "I know the need for learning in the operating room"; "I could share my learning experience in the operating room"; "I realize the concepts and methods of self-learning"; "I have been prepared for the surgical case"; "I know what I want to learn"; "I have good motivation to learn". A 6-point scale for responses ranged from far below required, moderately below required, mildly below required, reach required, above required, and far above required. The possible score ranges were from 6 to 36 and a higher score was indicative of better adult learning pattern. Reliability was measured using internal consistency and Cronbach's α was 0.93 (p< 0.001), revealing excellent reliability. The questionnaire was conducted after completing the perioperative training. Four levels of adult

learning pattern among study participants based on quartiles of ALQ would be defined. They were the highest, the high, the low, and the lowest level of adopting adult learning pattern.

The Clinical Core Competency Questionnaire (CCCQ) was a 6-item self-assessed questionnaire, developed and based on 6 ACGME clinical core competencies. It was administered using 6-point scales (1: far below required, 2: moderately below required, 3: mildly below required, 4: reach required, 5: above required, and 6: far above required). Possible total scores ranged from 6 to 36, with higher scores indicating better core surgical competency. Cronbach's α was equal to 0.93 (p <0.001), indicating excellent internal consistency. The questionnaires were learner-assessed before and after the 2-week rotation.

A question with three options (not at all, uncertain, and yes) was used to assess "interest to be a surgeon in the future", before and after the 2-week rotation.

## 2.3. Data management/statistical analyses

Continuous variables were expressed as mean ± SD and categorical variables were percentages. The significance of difference of continuous variables across groups was tested using independent $t$-test or one-way ANOVA, as appropriate. The difference in the distribution of categorical variables was tested using Chi-square test. The level of significance was set at $p < 0.05$ (two-tailed). Total score of ALQ would divided a list of numbers into quarters, and they were the highest level group ($\geq 75$ percentile, 75PR), high group (between 50PR and 75PR), low group (between 25PR and 50PR), and lowest (< 25PR) group.

## 3. Results

Altogether, 412 medical students (173 interns and 239 clerks) were enrolled in this study and 136 (33%) were female. Mean of ALQ was 30.8 (SD = 3.7). A lower quartile, median, and upper quartile of total ALQ were 28, 31, and 35 respectively, and four groups were created accordingly to represent the highest, the high, the low, and the lowest adult learning pattern adopted. Means of CCCQ were 23.7 (SD = 4.7) before and 30.0 (SD = 3.6) after perioperative training respectively. Paired Student $t$-test revealed there was a significant increase between pre- and post-training CCCQ (29.5, p <0.001).

Fig 1 illustrates the change of CCCQ from pre-training to post-training by four levels of adult learning pattern. The results show the increase of CCCQ across all four levels of adult learning, and they were from 21.5 (SD = 5.0) to 26.0 (SD = 2.7) for the lowest group, 23.3 (SD = 4.1) to 29.3 (SD = 2.0) for the low group, 24.4 (SD = 4.3) to 31.4 (SD = 2.0) for the high group, and 25.8 (SD = 4.3) to 33.9 (SD = 2.3) for the highest group. To examine if there was a difference of increment among four groups, we conducted a two-way repeated measure ANOVA (Table 1). The results revealed that there were differences in changes between pre- and post-CCCQ across four groups (interaction effect F = 13.0, p <0.001). That is, more increment of CCCQ was found among the highest adult learning group after complement of the structured perioperative training.

The interest to be a surgeon in the future was 22 (5.3%) *not at all*, 181 (43.9%) *uncertain*, and 209 (50.7%) *yes* before perioperative training; while 14 (3.4%) *not at all*, 144 (35.0%) *uncertain*, and 254 (61.7%) *yes* after perioperative training. Increased interest was defined with responses of "not at all" before training becoming "uncertain" or "yes"; or those with responses of "uncertain" before training becoming "yes". Totally, 61 (14.8%) students expressed increased interest to be a surgeon in the future (53 from undetermined to be positive, 7 from negative to be undetermined, and 1 from negative to be positive).

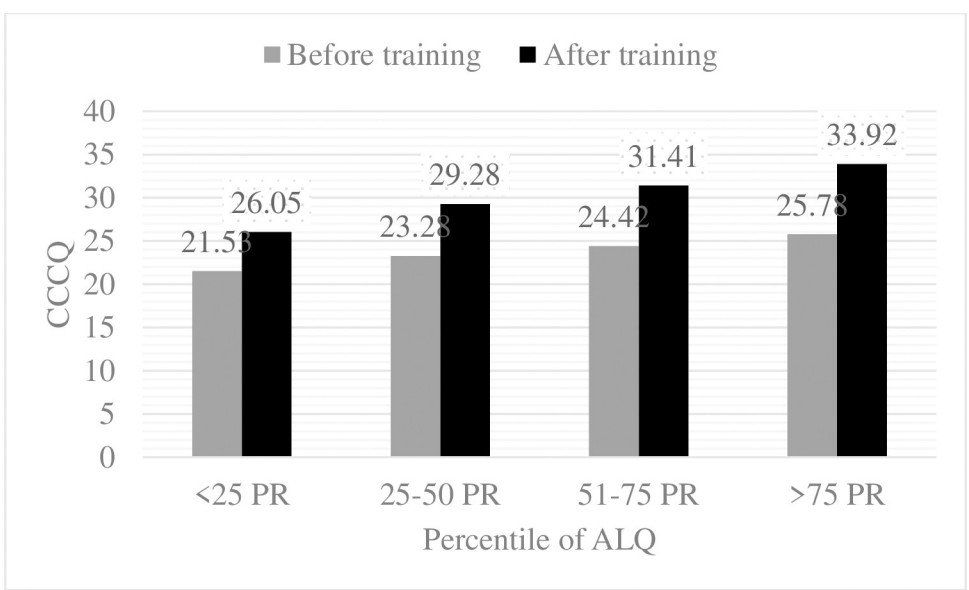

**Fig 1. Change of surgical competencies by 4 levels of adult learning pattern.** ALQ: Adult Learning Questionnaire; CCCQ: Clinical Core Competency Questionnaire; PR: percentile.

## 4. Discussion

The main findings of this study are as follow: (1) improvement of clinical core competency after completing the structural perioperative training; 2) improvement of clinical core competency was positively correlated with adult learning patterns; and 3) more learners expressed interest to be a surgeon in the future after perioperative training.

Roberts et al applied the briefing, intraoperative, and debriefing model for teaching in the operation room in 2009 [10], the modified structured perioperative teaching of this study, emphasizing the written record for debriefing, offers an effective learning and teaching model for perioperative training. The structured perioperative teaching model matched the learning styles with teaching styles by guided discovery learning [15]. Students' perceptions of perioperative teaching by faculty surgeons were strongly associated with the model. Postoperative feedback played an important role to encourage and motivate the learners to furthered self-

**Table 1. Pre- and post-perioperative training surgical competencies by quartile of adult learning among medical students.**

| ALQ | CCCQ | | Training effect | Adult learning effect | Interaction effect |
|---|---|---|---|---|---|
| | Pre-training | Post-training | | | |
| <25 PR | 21.53 (5.04) | 26.05 (2.71) | F = 955.76, p<0.001 | F = 78.84, p<0.001 | F = 13.0, p<0.001 |
| N = 102 | | | | | |
| 25~50 PR | 23.28 (4.14) | 29.28 (2.00) | | | |
| N = 115 | | | | | |
| 51~75 PR | 24.42 (4.29) | 31.41 (2.03) | | | |
| N = 106 | | | | | |
| >75 PR | 25.78 (4.29) | 33.92 (2.34) | | | |
| N = 89 | | | | | |

Presented with mean (SD)

ALQ: Adult Learning Questionnaire; CCCQ: Clinical Core Competency Questionnaire; PR: percentile

directed learning [21, 22]. Application of the clinical reasoning during debriefing feedback also effectively stimulate the learners to know what they have not known yet and what they need to know, and resulted in strong reflection and awareness of the importance of the adult learning. This inward awareness process made the students to actively learn better clinical competency related to the real patient, and adjusted and improved the self-weakness [11, 23]. The duty hours of surgical residents have been restricted by law since 2019 in Taiwan [24], in order to improve trainees' well-being and patient safety [25]. In the structural perioperative teaching, the learners were briefed with the preoperative preparation, operative indications, and objectives of surgery by the attending surgeon beginning from scrubbing time to start of operation. During the intraoperative teaching, surgical anatomy, instrument handling, surgical technique, details of procedure, independent practice, clinical reasoning, teamwork, and ethical issues would be intensively discussed between trainers and trainees. The possible complications, postoperative care, and prognosis would be included in the debriefing after surgery. Our findings suggested that the medical students perceived self-improvement in surgical competency with this structural perioperative training model.

There was a substantial number of medical students expressing increased interest to be a surgeon by nearly 15% after the perioperative training. Regardless of pre- or post-training, the interest rates of being a surgeon in this study were higher than those expected and compared to results of previous a study [26]. A possible explanation of this finding was that positive response to the research question did not necessarily exclude other career choices among the young medical students. Most medical students keep their career choice at a broad range during the beginning of the clinical learning stage; however, in terms of enrollment of more young surgeons, the findings of this study remain positive, as personal interest is usually the main factor influencing the final decision of career choice among junior doctors [27]. More importantly, this perioperative training not only provided an effective training program for medical students but possibly an efficient method to recruit more young doctors into the surgical field as well.

The more adult learning patterns medical students adapted, the higher the improvement of surgical competency after the structural perioperative training gained. Medical education has been designed to equip learners with the knowledge, clinical skills, and professionalism required to be a competent physician who delivers patient care with quality. As the perioperative training in this study required learners to experience more self-directed experiential learning, feedback and more teacher-learner-interaction, it could be expected that medical students with more adult learning patterns would have more effective learning outcomes represented by competency. The study emphasizes that definitely clear learning effects refers to so-called aesthetic education with the theoretic background of phenomenology of perception, meaning that learners pick up information via a variety of channels such as visual observation, aural listening, oral communication, cutaneous sensation, with olfactory and tactile sensation, and at the same time they combine all these perceptions, construct new meanings of learning objectives given during self-directed learning, thoughtful acquisition of protocols, and hands-on practice as well [28]. In brief, the learning outcome revealed an optimistic result, which was similar to our previous study [21].

Our study has some limitations. Firstly, it was a quasi-experiment design without a control group, and although there were two measurable outcomes, being clinical core competency and interest to be a surgeon merited the perioperative training, it was still difficult to definitively confirm the effectiveness of an intervention without a controlled group. Secondly, as study participants were enrolled from a single teaching hospital, the findings should be cautiously extrapolated to other populations. Thirdly, assessment in this study mainly relied on self-rating, possibly leading to some degree of rater bias. An objective assessment, particularly for

competency, would improve the study results. Finally, future studies with large study samples are necessary to confirm the findings.

## 5. Conclusions

A structural perioperative teaching model was re-developed, which included a written record for more focused feedback. The training improved clinical core competency and increased the interest of the students to be a future surgeon. The more adult learning patterns medical students develop, the more they will benefit from the training. Those whose interest in being surgeons increased improved their surgical competency more than their counterparts in this study.

## Supporting information

**S1 Data.**
(XLSX)

## Acknowledgments

The authors would like to acknowledge the contributions of all the participating students, colleagues, and research assistant Ching-Yu Hu who assisted with manuscript preparation.

## Author Contributions

**Conceptualization:** Nan-Chieh Chen, Yu-Tang Chang, Cheng-Sheng Chen.

**Data curation:** Po-Chih Chang, Cheng-Sheng Chen, Chung-Sheng Lai.

**Formal analysis:** Nan-Chieh Chen.

**Investigation:** Cheng-Sheng Chen.

**Methodology:** Cheng-Sheng Chen.

**Supervision:** Chung-Sheng Lai.

**Writing – original draft:** Nan-Chieh Chen, Yu-Tang Chang, Po-Chih Chang, Cheng-Sheng Chen, Chung-Sheng Lai.

**Writing – review & editing:** Yu-Tang Chang, Cheng-Sheng Chen, Chung-Sheng Lai.

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
