## [Decision Letter · Decision Letter 0]

12 Nov 2021

PONE-D-21-31480Learning Outcomes of Structured Perioperative Teaching Based on Adult LearningPLOS ONE

Dear Dr. Chen,

Thank you for submitting your manuscript to PLOS ONE. After careful consideration, we feel that it has merit but does not fully meet PLOS ONE’s publication criteria as it currently stands. Therefore, we invite you to submit a revised version of the manuscript that addresses the points raised during the review process. Both reviewers  made an effort to give recommendations to improve your manuscript.You are kindly invited to seek help of a native speaker. Reviewer 1 summarizes details you should follow for the revision.

We look forward to receiving your revised manuscript.

Kind regards,

Hans-Peter Simmen, M.D., Professor of Surgery

Academic Editor

PLOS ONE

Journal Requirements:

4. We note you have included a table to which you do not refer in the text of your manuscript. Please ensure that you refer to Table 1 in your text; if accepted, production will need this reference to link the reader to the Table.

Reviewers' comments:

Reviewer's Responses to Questions

**Comments to the Author**

1. Is the manuscript technically sound, and do the data support the conclusions?

Reviewer #1: Yes

2. Has the statistical analysis been performed appropriately and rigorously? 

Reviewer #1: Yes

3. Have the authors made all data underlying the findings in their manuscript fully available?

Reviewer #1: Yes

4. Is the manuscript presented in an intelligible fashion and written in standard English?

Reviewer #1: No

5. Review Comments to the Author

Reviewer #1: • “shortage of systemic didactics in perioperative training”, that is quite a general assumption. Can u provide some references?

• , which should indeed provide students with more information about knowledge and skills for operating so that more interest and intention to being surgeons might be aroused.

o Rephrase please.

• “. However, to improve the working environment by reducing working hours might lead to inadequate training in the operating room.”

o What are the standard working hours and what are the average real working hours? What about lack of work quality due to overworked stuff?

• It is necessary to develop an effective learning environment to balance learning and teaching effectiveness and working load.

o How does reduce the work load?

• “Traditional curriculum design and didactic approach in surgical education tend to be teaching strategies with little structure where the content of teaching is not well organized”

o That’s also a really general assumption which would need further explanation of the implemented system and the lack of it.

• Traditional curriculum design and didactic approach in surgical education tend to be teaching strategies with little structure where the content of teaching is not well organized. and the learners play the role more as absolute passive observers than the ones with active participation, even under a highly tense atmosphere in the operating room.

o Rephrase and check gramma.

o What do u mean with a “tense atmosphere”?

• “Consequently, a disparity might exists between learners and teaching faculty in terms of perioperative teaching”

o I can’t find prove of that statement and the references included.

• “Adult learning itself is distinguished by having been put forward from childhood learning, in which the learners develop self-directed learning both in learning goals and strategies. The theory points out that adults learn best when they clearly know their motivation for learning.

o Rephrase and inset citation

• This study was conducted in a medical center in Taiwan

o What kind of center? Which level?

• The intention was that the written record would allow learners to become aware of adult learning principles , then the teacher would closely review the written record and give direct feedback to the learners immediately after operation Application of the problem centered clinical reasoning through Questions and Answers (Q &A) interaction related to the patients’ condition could evoke the awareness of learning needs. Therefore, the structural learning model with written record designed in the teaching study based on adult learning pattern and clinical reasoning of teaching and learning.

o Rephrase

• “ The questionnaire was conducted after completing the perioperative training”

o Where is the control group in this setting when u only asked participant’s after the training?

• ; however, possibly at the expense of training effectiveness and quality.

o this is a very bold and generalized statement. Please discuss here the advantages and disadvantages of regulated working hours and regarding teaching, time for self-study and preparation and overwork.

• In the structural perioperative teaching, the learners were briefed with the preoperative preparation, operative indications, and objectives of surgery by the attending surgeon at the scrubbing time

o Scrubbing time is usually not more than 3 minutes. Please discuss if that is a appropriate timeframe for teaching.

6. PLOS authors have the option to publish the peer review history of their article (what does this mean?). If published, this will include your full peer review and any attached files.

Reviewer #1: No

---

## [Author Response · Author response to Decision Letter 0]

1 Jan 2022

Responses to reviewer’s Questions

Reviewer #1: 

1. “shortage of systemic didactics in perioperative training”, that is quite a general assumption. Can u provide some references?

Response: Thanks for the suggestion. Two references (ref 1 and 2) suggested a shortage of systemic didactics in perioperative training have been added.

2. • , which should indeed provide students with more information about knowledge and skills for operating so that more interest and intention to being surgeons might be aroused.

o Rephrase please.

Response: Thanks for the suggestion. The sentence has been rephrased. Please see the first paragraph of the introduction section.

3. • “. However, to improve the working environment by reducing working hours might lead to inadequate training in the operating room.”

o What are the standard working hours and what are the average real working hours? What about lack of work quality due to overworked stuff?

Response: The government regulated the maximum working hours of 80 hours per week, however, the real working hours possibly exceeded 110 hours per week before the law establishment. To prevent lack of work quality due to overworked stuff, the government has set the working hours restriction as one of teaching hospital accreditation standards. 

4. • It is necessary to develop an effective learning environment to balance learning and teaching effectiveness and working load.

o How does reduce the work load?

Response: Work loading could be probably reduced by duty hour restriction. The government has required all hospitals to follow the law in our country. 

5. • “Traditional curriculum design and didactic approach in surgical education tend to be teaching strategies with little structure where the content of teaching is not well organized”

o That’s also a really general assumption which would need further explanation of the implemented system and the lack of it.

Response: Thanks for the suggestion. The sentence has been rephrased and a few references cited to support the background. Please see the second paragraph of the introduction section in the revised manuscript.

6. • Traditional curriculum design and didactic approach in surgical education tend to be teaching strategies with little structure where the content of teaching is not well organized. and the learners play the role more as absolute passive observers than the ones with active participation, even under a highly tense atmosphere in the operating room.

o Rephrase and check gramma.

o What do u mean with a “tense atmosphere”?

Response: We rethink the term “tense atmosphere” which seems to be subjective and misleading. The phrase has been deleted in the revised manuscript. The grammatical error of the whole sentence has been also corrected. Thank you for the suggestion. 

7. • “Consequently, a disparity might exists between learners and teaching faculty in terms of perioperative teaching”

o I can’t find prove of that statement and the references included.

Response: The sentence has been rewritten more clearly. A reference has been removed because it could not be found in the online literature database has been removed. Thank you for the suggestion.

8. • “Adult learning itself is distinguished by having been put forward from childhood learning, in which the learners develop self-directed learning both in learning goals and strategies. The theory points out that adults learn best when they clearly know their motivation for learning.

o Rephrase and inset citation

Response: Thanks for the suggestion. The sentence has been rephrased and one reference (ref 12) has been cited in the revised manuscript.

9. • This study was conducted in a medical center in Taiwan

o What kind of center? Which level?

Response: It should be a university teaching hospital. We had made it clearer in the revised manuscript.

10. • The intention was that the written record would allow learners to become aware of adult learning principles , then the teacher would closely review the written record and give direct feedback to the learners immediately after operation Application of the problem centered clinical reasoning through Questions and Answers (Q &A) interaction related to the patients’ condition could evoke the awareness of learning needs. Therefore, the structural learning model with written record designed in the teaching study based on adult learning pattern and clinical reasoning of teaching and learning.

o Rephrase

Response: Thanks for the suggestion. The sentences have been rephrased. Please see the last paragraph of the 2.1 section. 

11. • “ The questionnaire was conducted after completing the perioperative training”

o Where is the control group in this setting when u only asked participant’s after the training?

Response: This medical education study was a quasi-experiment design without a control group. The limitation of such study design has been clearly mentioned in the discussion section. Please see the last paragraph of the discussion section. 

12. • ; however, possibly at the expense of training effectiveness and quality.

o this is a very bold and generalized statement. Please discuss here the advantages and disadvantages of regulated working hours and regarding teaching, time for self-study and preparation and overwork.

Response: Thanks for the suggestion. This was indeed a bold and generalized statement and easily misunderstood. As the duty hours policy is not the main point of this study, we decided to delete the part of the discussion. 

13. • In the structural perioperative teaching, the learners were briefed with the preoperative preparation, operative indications, and objectives of surgery by the attending surgeon at the scrubbing time

o Scrubbing time is usually not more than 3 minutes. Please discuss if that is a appropriate timeframe for teaching.

Response: Thanks for the correction. “At the scrubbing time” did not exactly describe what it was. Instead, the briefing started at the scrubbing time and continued after scrubbing to the beginning of the operation. It usually took 10 minutes. Please see the second paragraph of the discussion section.

---

## [Editor Report · Decision Letter 1]

7 Jan 2022

Learning Outcomes of Structured Perioperative Teaching Based on Adult Learning

PONE-D-21-31480R1

Dear Dr. Chen,

We’re pleased to inform you that your manuscript has been judged scientifically suitable for publication and will be formally accepted for publication once it meets all outstanding technical requirements.

Kind regards,

Hans-Peter Simmen, M.D., Professor of Surgery

Academic Editor

PLOS ONE
---

## [Editor Report · Acceptance letter]

13 Jan 2022

PONE-D-21-31480R1 

Learning Outcomes of Structured Perioperative Teaching Based on Adult Learning 

Dear Dr. Chen:

I'm pleased to inform you that your manuscript has been deemed suitable for publication in PLOS ONE. Congratulations! Your manuscript is now with our production department. 

Kind regards, 

on behalf of

Dr. Hans-Peter Simmen 

Academic Editor

PLOS ONE